# A Methodological Approach for Implementing an Integrated Multimorbidity Care Model: Results from the Pre-Implementation Stage of Joint Action CHRODIS-PLUS

**DOI:** 10.3390/ijerph16245044

**Published:** 2019-12-11

**Authors:** Katie Palmer, Angelo Carfì, Carmen Angioletti, Antonella Di Paola, Rokas Navickas, Laimis Dambrauskas, Elena Jureviciene, Maria João Forjaz, Carmen Rodriguez-Blazquez, Alexandra Prados-Torres, Antonio Gimeno-Miguel, Mabel Cano-del Pozo, María Bestué-Cardiel, Francisca Leiva-Fernández, Elisa Poses Ferrer, Ana M Carriazo, Carmen Lama, Rafael Rodríguez-Acuña, Inmaculada Cosano, Juan José Bedoya-Belmonte, Ida Liseckiene, Mirca Barbolini, Jon Txarramendieta, Esteban de Manuel Keenoy, Ane Fullaondo, Mieke Rijken, Graziano Onder

**Affiliations:** 1Department of Internal Medicine and Geriatrics, Università Cattolica del Sacro Cuore, 00136 Rome, Italy; antonella894@hotmail.it; 2Centro di Medicina dell’Invecchiamento, Fondazione Policlinico Universitario “A. Gemelli” IRCCS, 00136 Rome, Italy; 3Department of Internal Medicine and Geriatrics, Università Cattolica del Sacro Cuore and Centro di Medicina dell’Invecchiamento, Fondazione Policlinico Universitario “A. Gemelli” IRCCS, 00136 Rome, Italy; carmen.angioletti92@gmail.com; 4Faculty of Medicine, Vilnius University, LT-03101 Vilnius, Lithuania; rokas.navickas@santa.lt (R.N.); Laimis.Dambrauskas@santa.lt (L.D.); Elena.Jureviciene@santa.lt (E.J.); 5Department of Biomedical Research, Vilnius University Hospital Santaros Klinikos, LT-08661 Vilnius, Lithuania; 6National School of Public Health and REDISSEC, Carlos III Institute of Health, ES-28029 Madrid, Spain; jforjaz@isciii.es; 7National Centre of Epidemiology and CIBERNED, Carlos III Institute of Health, ES-28029 Madrid, Spain; crodb@isciii.es; 8EpiChron Research Group, Aragon Health Sciences Institute (IACS), IIS Aragón, Miguel Servet University Hospital, REDISSEC, 50009 Zaragoza, Spain; sprados.iacs@aragon.es (A.P.-T.); agimenomi.iacs@aragon.es (A.G.-M.); 9General Directorate of Healthcare, Health Department, 50017 Zaragoza, Spain; micano@aragon.es (M.C.-d.P.); mbestue@salud.aragon.es (M.B.-C.); 10Málaga-Guadalhorce Primary Care Teaching Unit, IBIMA, Andalusian Health Service, 29009 Málaga, Spain; francisca.leiva.sspa@juntadeandalucia.es; 11Agency for Health Quality and Assessment of Catalonia (AQuAS), Government of Catalonia, 08005 Barcelona, Spain; eposes@gencat.cat; 12Regional Ministry of Health and Families of Andalusia (CSFJA), E-41020 Seville, Spain; anam.carriazo@juntadeandalucia.es (A.M.C.); carmenm.lama@juntadeandalucia.es (C.L.); 13Andalusian Public Foundation Progress and Health (FPS), E-41092 Seville, Spain; rafael.rodriguez.acuna@juntadeandalucia.es; 14 San José de la Rinconada-Los Carteros Primary Care Center, Andalusian Health Service (Servicio Andaluz de Salud, SAS), E-41300 Seville, Spain; 15 Tiro de Pichón Primary Care Center, Andalusian Health Service (Servicio Andaluz de Salud, SAS), E-29006 Málaga, Spain; bedoyabelmonte@gmail.com; 16Faculty of Medicine, Lithuanian University of Health Sciences, LT-44307 Kaunas, Lithuania; Ida.Liseckiene@kaunoklinikos.lt; 17European Commission (DG Santè), 41225 Modena, Italy; mircabarbolini@gmail.com; 18Kronikgune Institute for Health Services Research, 48902 Basque Country, Spain; jtxarramendieta@kronikgune.org (J.T.); edemanuel@kronikgune.org (E.d.M.K.); afullaondo@kronikgune.org (A.F.); 19Nivel (Netherlands Institute for Health Services Research), 3513 CR Utrecht, The Netherlands; m.rijken@nivel.nl; 20Department of Health and Social Management, University of Eastern Finland, FI-70210 Kuopio, Finland; 21Department of Cardiovascular, Metabolic and Aging Diseases, Istituto Superiore di Sanità, 00161 Rome, Italy

**Keywords:** multimorbidity, chronic disease, non-communicable diseases, integrated care, care model, Europe, care manager, individualized care plans, comprehensive assessment

## Abstract

Patients with multimorbidity (defined as the co-occurrence of multiple chronic diseases) frequently experience fragmented care, which increases the risk of negative outcomes. A recently proposed Integrated Multimorbidity Care Model aims to overcome many issues related to fragmented care. In the context of Joint Action CHRODIS-PLUS, an implementation methodology was developed for the care model, which is being piloted in five sites. We aim to (1) explain the methodology used to implement the care model and (2) describe how the pilot sites have adapted and applied the proposed methodology. The model is being implemented in Spain (Andalusia and Aragon), Lithuania (Vilnius and Kaunas), and Italy (Rome). Local implementation working groups at each site adapted the model to local needs, goals, and resources using the same methodological steps: (1) Scope analysis; (2) situation analysis—“strengths, weaknesses, opportunities, threats” (SWOT) analysis; (3) development and improvement of implementation methodology; and (4) final development of an action plan. This common implementation strategy shows how care models can be adapted according to local and regional specificities. Analysis of the common key outcome indicators at the post-implementation phase will help to demonstrate the clinical effectiveness, as well as highlight any difficulties in adapting a common Integrated Multimorbidity Care Model in different countries and clinical settings.

## 1. Introduction

### 1.1. The Challenge of Multimorbidity

Multimorbidity is the co-occurrence of multiple chronic diseases or conditions in a single individual. It has been described as the most common chronic condition, as it has a high prevalence, especially in older individuals, where it affects more than 60% of people aged 65 or over [1]. Some multimorbidity patients can be complex, particularly because they are more likely to have problems with mobility, self-care, and daily functioning than patients with one chronic disease, as well as cognitive impairment and frailty [2]. This often results in more challenging healthcare treatment. Many healthcare systems still focus on a traditional disease-oriented approach. Consequently, multimorbid patients frequently experience fragmented care [3,4] and receive complex drug regimens and polypharmacy, which increase the risk of inappropriate prescribing, adverse drug reactions, and poor medication adherence [5].

[JA CHRODIS PLUS]—the Joint Action (JA) CHRODIS-PLUS: Implementing good practices for chronic diseases (JA-CHRODIS-PLUS)

[JA CHRODIS]—the Joint Action on Chronic Diseases and Promoting Healthy Ageing across the Life Cycle (JA-CHRODIS)

### 1.2. The Integrated Multimorbidity Care Model

An Integrated Multimorbidity Care Model [6] was recently proposed, which aims to overcome many of the issues related to fragmented care. The model was developed as part of the Joint Action (JA) on Chronic Diseases and Promoting Healthy Ageing across the Life Cycle (CHRODIS) [7] and focuses on several limitations currently faced in the treatment of multimorbid patients. It recognizes that fragmented care may be due to a lack of integration between primary and hospital care services as well as between healthcare professionals from different specialties or disciplines. Currently, although many healthcare professionals are well trained to manage single chronic diseases by following official clinical guidelines for specific diseases, they are not specifically trained to handle patients with multimorbidity. They also may be inexperienced in terms of adopting patient-centered care and shared-decision making that takes the patient’s preferences, needs, and expectations into account. The Integrated Multimorbidity Care Model, therefore, proposes 16 components for the care and treatment of multimorbid patients. These components are categorized into five domains: Delivery of care; decision support; self-management support; information systems and technology; and social and community resources. Much of the basic structure of the model is based on Wager et al.’s Chronic Care Model [8,9] and was adapted to the specific needs of multimorbid patients, based on scientific evidence combined with the opinions of international experts. However, the model has not been tested in real-life clinical practice until now. In 2016 a systematic review [10] found only 19 publications in the scientific literature that assessed integrated care models for multimorbidity and only one of these was from Europe. Palmer et al. [6] highlighted that the applicability of the JA CHRODIS Integrated Multimorbidity Care Model will depend on type of health and care service system of the country and that the guidelines should be interpreted and applied according to the specific setting. The current paper describes the development of a methodology to operationalize principles described in the model and its application to various clinical settings in different European countries.

### 1.3. Pilot Implementation of the Integrated Multimorbidity Care Model: Joint Action CHRODIS-PLUS

The JA CHRODIS-PLUS on Implementing Good Practices for Chronic Diseases [11] is a three-year project funded by the European Union that aims to support European member states through the implementation of cross-national policies and practices with demonstrated success to reduce the burden of chronic disease identified in JA CHRODIS. One of the main objectives of JA CHRODIS-PLUS is to develop a methodology to implement the Integrated Multimorbidity Care Model. This led to the definition of a framework for the care of patients with multimorbidity that could potentially be adapted and applied in local practices in Europe [6]. Such a methodology is then applied in pilot sites and its effectiveness on clinical and process outcomes is tested.

### 1.4. Aims and Objectives

In the context of JA CHRODIS-PLUS, an implementation methodology was developed, which is currently being piloted in five sites in Spain, Lithuania, and Italy, where the model is adapted and implemented according to local practices. The aim of the current paper is to (i) explain the methodology used to implement the Integrated Multimorbidity Care Model and (ii) describe the five pilot sites and how they have adapted and applied the proposed methodology for local implementation.

## 2. Methods

### 2.1. Survey to Assess Characteristics of the Pilot Sites

At the start of the project, a survey was designed to assess characteristics of the five sites that would be participating in the implementation. JA CHRODIS-PLUS partners designed a questionnaire to collect information about the organizations and their planned care model programs, across six dimensions: (1) General information; (2) delivery of care and decision support; (3) patient self-management; (4) eHealth; (5) community resources; and (6) practice/program assessment. After the development of the questionnaire, an online version was made accessible to partners. The survey was used to identify common characteristic of the five pilot sites as well as to explore their differences.

### 2.2. Patient Risk Stratification Strategies

Pilot sites were asked to adopt a risk stratification process to ensure that care coordination would focus on patients who would benefit the most, thus maximizing the impact on both quality and costs. Risk stratification is defined as a systematic process to target, identify, and select patients who are at risk of poorer health outcomes, and who are expected to benefit most from an intervention [12,13,14]. The process groups the population according to different risk levels and needs based on how likely people are to use services and resources and also helps identify practices where improvement is necessary.

Each of the five pilot sites first defined a target cohort of persons who were at risk of poorer health outcomes and considered a priority for targeting with different or additional interventions. They identified individuals within the target cohort by searching databases that routinely collect information on clinical or demographic data. Individuals were then selected by healthcare professionals according to their need for the integrated care intervention.

### 2.3. Implementation Strategy

A common implementation strategy was developed for all the implementation pilot sites, which aimed to provide guidelines to facilitate the uptake of routine good practices, policies, and tools. This implementation strategy was designed by JA CHRODIS-PLUS coordinators, partners, and other dedicated experts. 

As a first step, each of the five sites established a local implementation working group comprised of beneficiaries, collaborative care providers, and local stakeholders. Although the composition of the local implementation working group could differ between sites, all of them had to include a core set of persons in the team, specifically: Organizer, experts, decision makers, front line stakeholders, and implementers (see Table 1). The working groups involved periodic face-to-face meetings (when this was not possible, online meetings were held) of 2–3 h of duration with specific tasks for each meeting: (1) Scope analysis; (2) situation analysis—“strengths, weaknesses, opportunities, threats” (SWOT); (3) development and improvement of methodology; and (4) final development of the pilot action plan (see Figure 1).

#### 2.3.1. Implementation Strategy Step 1: Scope Analysis

During the scope analysis, each local implementation working group selected the specific features or elements of their planned intervention, which were identified according to their health context and local needs, interests, and capabilities. A structured group discussion was used. Although the criteria for defining the scope could differ between sites, they generally followed five steps: (1) Identify and describe the problem/challenge; (2) describe the general purpose of the intervention; (3) describe the target population; (4) analyze the intervention’s components and identify the central features that are essential to achieve the desired results; and (5) select the components from the Integrated Multimorbidity Care Model that will be locally implemented.

#### 2.3.2. Implementation Strategy Step 2: SWOT Analysis

Situation analysis—“strengths, weaknesses, opportunities, threats” (SWOT)—was used to identify the respective organizations’ internal strengths and weaknesses, as well as external opportunities for, and threats to, implementing the interventions based on the selected model elements. SWOT is designed to help with both strategic planning and decision making in relation to the planned intervention. SWOT was chosen as a tool because it is a structured method that is comparable. This allowed us to compare the different analyses from the five sites.

During the SWOT analysis, the working groups considered the strengths, weaknesses, opportunities, and threats to their proposed Integrated Multimorbidity Care Model across five dimensions: (1) Sustainability; (2) organization; (3) empowerment; (4) communication; and (5) monitoring and evaluation. A template was devised to facilitate discussion. All five sites prepared a matrix that presented the most important strengths, weaknesses, opportunities, and threats for their organization, with an overview of major issues, priorities, and strategic actions needed in relation to their planned intervention.

#### 2.3.3. Implementation Strategy Steps 3 and 4: Development and Improvement of Methodology and Final Development of Action Plans

The methodology was developed and improved by the five local working groups during the face-to-face meetings, leading to the development of an action plan, which provides a concrete set of steps and activities that need to be carried out in order to implement their respective care interventions. An adapted version of the iterative cyclic nature of “collaborative methodology” [15] was used for drafting the local action plans. According to this methodology, the working groups addressed three main questions: (1) What are we trying to accomplish? (2) What changes can we make that will result in a successful implementation of the Integrated Multimorbidity Care Model as well as improvement? (3) How will we know that a change is an improvement? These questions were used to develop a concrete action plan, which was devised in five steps (see Table 2).

## 3. Results

The Integrated Multimorbidity Care Model is being implemented in five pilot sites from Spain (the Andalusian Health System and the Aragon Health System), Lithuania (Vilnius University Hospital Santaros Klinikos, VULSK, Vilnius and Hospital of Lithuanian University of Health Sciences Kauno Klinikos, Kauno Klinikos, Kaunas), and Italy (Università Cattolica del Sacro Cuore (UCSC), Rome). As described in the methods section, a survey was carried out at the start of the project to identify characteristics of the participating centers before the implementation of the Integrated Multimorbidity Care Model. Results of the survey revealed some common goals for the five pilot sites, such as to increase multidisciplinary collaboration, promote evidence-based practice, and reduce inequalities in access to care and support services. A summary of some of the characteristics of the sites is illustrated in Table 3. All five pilot sites include a six-month run-in period (patient recruitment), followed by a 12-month implementation period. Key indicators are measured at the end of the implementation. JA CHRODIS-PLUS is a three-year project and the timescale of the interventions were chosen in order to allow sufficient time for preparation, application of the intervention, and reporting. Most of the implementers considered it important to involve general practitioners and nurses in delivering care to patients. Indeed, the majority of patients are being identified via primary care settings. In all cases, the main care providers are either general practitioner physicians or nurses (or they are involved in the multidisciplinary meetings). Case managers are appointed in the majority of interventions (usually a physician) and many also include a social worker as part of the core multidisciplinary team. All five sites report that their patients will undergo comprehensive assessment at the start and end of the integrated care process, but few include a regular periodic assessment in-between. Most of the programs reported some key common characteristics of the intervention and services, patient education, follow-up visits, and referrals between medical specialties have been reported by all five sites, and clinical (diagnostic/monitoring) tests in all but one. However, other characteristics of the intervention and services differ somewhat between settings.

Most sites are using technology in their interventions. For example, four of the five sites offer eHealth services and half of the multidisciplinary team meetings are conducted virtually. All five sites reported using digital health care communication tools; these are mostly e-referral but there are also other aspects like virtual conferences with patients and online appointment schedules. Three-quarters of the sites have electronic systems for registering/monitoring care processes and all use electronic health records. However, currently none of the programs use electronic decision support systems. The survey also highlighted some noticeable absences, especially in terms of community and social resources. In fact, only one site is directly supporting patients in accessing community and social resources.

### 3.1. Components in the Planned Interventions

The five sites were required to implement at least one component from the 2018 Multimorbidity Care Model proposed by JA-CHRODIS [6], which proposed 16 components. Table 4 describes which elements were chosen to be included in each site’s intervention. Kauno Klinikos are implementing 13 of the 16 components and three sites (Kauno Klinikos, UCSC-Rome, and VULSK) are including components from all of the five domains. The Andalusian Health System’ intervention focuses only on the “individualized care plan” component; other components are already in place in this region. Most sites (four out of five) include regular, comprehensive assessment of patients, a multidisciplinary team, a case manager, individualized care plans, and shared decision making between patients and care providers. Only one site (Aragon Health System) is providing training to care providers on supporting patient self-management, while another (UCSC-Rome) includes patient operated technologies that allow patients to send information to their care providers. In the Andalusian Health System, the Observatory of Innovative Practices for Complex Chronic Diseases Management (OPIMEC) [16] already provides online knowledge exchange and training to professionals involved in the management and treatment of complex chronic patients on a regular basis.

### 3.2. Description of Pilot Sites

The Andalusian Health System: “Implementation of a ‘personalized action plan’ within the strategy and comprehensive plan for complex chronic patients”.

The “Consejería de Salud y Familias de la Junta de Andalucía” is implementing their Integrated Multimorbidity Care Model in primary care centers all over this region of Spain. The implementation is linked with the healthcare strategy for complex chronic patients, within the framework of the Andalusian comprehensive healthcare plan for patients with chronic diseases. The plan focuses on enhancing community care (primary healthcare), intra-level coordination, and continuity of care (using a liaison nurse). The objectives of the care model are to: increase multidisciplinary collaboration, improve patients and informal careers involvement, improve functional status, decrease and delay complications, to reduce inequalities in access to care and support services, and reduce hospital admissions and acute care visits. The intervention targets one component of the Integrated Multimorbidity Care Model (see Table 4), namely individualized care plans, although other components of the model are already in place. The specific aim of the intervention is to assess the influence of the systematized application of individualized and comprehensive care plans to complex chronic patients (patients with chronic severe health problems, multimorbidity, and polypharmacy). All complex chronic patients with individualized care plans started and delivered between December 2018 and February 2019 were selected as the target population and will be followed for one year. Regular training of healthcare professionals is provided through the OPIMEC platform [16].

The Aragon Health System: “Aragon primary care”.

The model is being implemented in a total of 13 primary health care centers in the Aragon Health System. A total of 43 healthcare professionals (21 general practitioners, 18 primary care nurses, 2 internal medicine specialists, and 2 internal medicine nurses) from 13 health care centers and 1 hospital, with a long professional experience, have been trained in multimorbidity through the eMulti-PAP course developed within the framework of the Multi-PAP randomized control trial [17]. A total of 291 high risk multimorbid patients from their respective practices have been selected and included in the piloting. The main aim is to examine the feasibility of implementing this type of intervention in a real context and to decrease the impact of multimorbidity on health outcomes in patients aged 65 years and over with multimorbidity (≥3 chronic diseases) and polypharmacy (≥5 drugs). The main objectives of the care model are to: promote evidence-based practice, reduce inequalities in access to care and support services, prevent or reduce misuse of services, increase multidisciplinary collaboration, and decrease morbidity. The intervention targets eleven components of the Integrated Multimorbidity Care Model (see Table 4) from all domains. They include training for healthcare providers, appointment of a case manager, use of individualized care plans, development of a virtual inter-consultation system, and supporting access to community resources.

UCSC-Rome outpatient clinic in the Catholic University of the Sacred Heart, Rome, Italy: “Multimorbidity care model in elders with dementia and adults with intellectual disability”.

UCSC-Rome are implementing their model in a national health service run tertiary care hospital (Università Cattolica del Sacro Cuore, Fondazione Policlinico Universitario Agostino Gemelli) in Rome, Italy. The clinical government unit is mainly involved in this project together with the center for ageing medicine (Centro Medicina dell’Invecchiamento). The care model is carried out in a day hospital and focuses on ageing, frail patients with intellectual disability, comorbidity/multimorbidity, and cognitive impairment. The aim of the intervention is to improve coordination and provide patients with a reference care provider as well as to increase accessibility of care through a Technocare service and enhance self-management through patient-operated technology. The main objectives of the care model are to: improve professional knowledge on multimorbidity, reduce inequalities in access to care and support services, improve accessibility of services, improve care coordination and integration of different units (within the organization), increase multidisciplinary collaboration, identifying target group patients, improve patient and informal career involvement, and reduce hospital admissions and acute care visits. The intervention will target nine components of the Integrated Multimorbidity Care Model (see Table 4), from all five domains.

Kauno Klinikos: Kauno clinics primary healthcare center and Kaltinenai primary healthcare center, Kaunas, Lithuania.

Kauno Klinikos is implementing the care model in the family medicine department of a tertiary university clinic located in the second largest Lithuanian city. It provides all scope of primary care services and is in close relation with other health sectors: secondary and tertiary as well. The target population includes patients with multimorbidity aged 45–70, identified by GPs. The aim of the intervention is to test the Integrated Multimorbidity Care Model patients in Lithuania to provide better care for multimorbid patients and improve their quality of life, decrease the number of potentially avoidable hospitalizations and readmissions, to elaborate economical evaluation of the expenditure for the multimorbid patients. The main objectives are to: reduce adverse outcomes related to the presence of multiple diseases and the risk of drug-drug interactions by elaborating individualized integrated care plans, optimize treatment, maintenance, and healthcare resources by coordinating individualized integrated care plans; maximize outcomes and increase continuity of care while decreasing fragmentation and optimizing access to care and services through a case manager, who will intermediate between a patient and various members of the multidisciplinary team; provide doctor-to-doctor decision support in situations where further clinical support or knowledge is needed outside of the core team through a consultation system to be advised by professional experts; and improve the patient‘s access to community resources, formal care, and patient associations, support groups, and psychosocial support by providing multidisciplinary care both in terms of different levels of the healthcare profession (nurses, physicians, physiotherapists, social workers etc.), and different disease specializations. The intervention targets 13 components of the Integrated Multimorbidity Care Model (see Table 4), from all five domains. 

VULSK family medicine center primary care setting at Vilnius University Hospital Santaros Klinikos, Vilnius, Lithuania: “Family medicine center, primary care”.

VULSK is implementing the care model in primary care but is also expanding beyond the primary care setting, to include the secondary and tertiary care physicians, aiming to create teams who manage the patient. It is aimed at multimorbid patients attending primary care settings. The main objective of the program is to promote evidence-based practice to primary care multimorbid patients with the aim to improve their quality of life, decrease the number of potentially avoidable hospitalizations and readmissions, and elaborate economical evaluation of the expenditure for the multimorbid patients. The intervention targets 10 components of the Integrated Multimorbidity Care Model (see Table 4) from all five domains. In particular, it includes all components from the delivery of care model and the decision support components. The specific aims of the intervention include to: (1) Reduce adverse outcomes related to the presence of multiple diseases and the risk of drug-drug interactions by elaborating individualized integrated care plans; (2) optimize treatment, maintenance, and healthcare resources by coordinating individualized integrated care plan; (3) maximize outcomes and increase continuity of care while decreasing fragmentation and optimizing access to care and services through a case manager; (4) provide doctor-to-doctor decision support in situations where further clinical support or knowledge is needed outside of the core team through a consultation system of professional experts; (5) improve the patient‘s access to community resources, formal care, and patient associations, support groups, and psychosocial support by providing multidisciplinary care both in terms of different levels of the healthcare profession and different disease specializations.

### 3.3. Key Performance Indicators

During development of the action plans, each pilot site defined key performance indictors to measure the success of the respective interventions. A common approach was chosen for assessing the impact of the interventions that consisted both of a quantitative and qualitative analysis. The specific key performance indicators for each site are described in Table 5. There were some common indicators between sites, particularly Kauno Klinikos and VULSK, who defined a similar set of indicators. 

The Assessment of Chronic Illness Care (ACIC) [18] questionnaire was chosen as an appropriate quantitative measure. ACIC is responsive to changes that care teams make in their healthcare systems and correlates well with other measures of productivity and system change. It consists of six elements that were proposed in the Chronic Care Model, namely; health care organization; community linkages; self-management support; decision support; delivery system design; and clinical information systems. The Patient Assessment of Care for Chronic Conditions+ (PACIC+) [19] was also selected for quantitative measuring outcomes of the interventions. This tool measures specific actions or qualities of care that patients report they have experienced during the intervention. The actions are congruent with the Chronic CareModel and consist of 26 items. Both instruments will be collected and analyzed pre- and post-implementation.

## 4. Discussion

In this article we describe the methods of the pre-implementation phase of five pilot sites who are implementing an Integrated Multimorbidity Care Model as part of JA CHRODIS-PLUS. Key stages of the pre-implementation phase include the establishment of a local implementation working group, patient risk stratification, scope analysis, SWOT analysis, development and improvement of methodology, and final development of the pilot action plan. Characteristics of the five pilot sites and their specific interventions are described in terms of the intervention’s main aim and objectives, settings and patients, and key quality and performance indicators. This common implementation strategy serves to show how care models can be adapted according to local and regional specificities.

There are several strengths of the methodology. We used a process that followed a standardized procedure that could be adapted according to the local site’s needs, capabilities, and characteristics. The process was developed so that each site followed the same methodology in order to create standardized implementation package that can be practically applied in different European and clinical settings. Each of the five sites participated in regular joint meetings (usually virtually) with the Work Package coordinators to compare strategies and identify any unclarities in the methodology. In addition, we used tools such as SWOT analysis that can provide comparable information that identify differences and similarities across sites.

There are potential limitations to the implementation process described herein. First, the implementation process was set at 18 months in order to fit within the timeframe of the three-year JA CHRODIS-PLUS project. Therefore, various aspects such as the selected key indicators might reflect this relatively short intervention period. Longer interventions might have the potential to assess other relevant health related indicators such as mortality, change in frailty status, or long-term functional status, as well as economical and cost-effectives data. In addition, some of the key indicators need to be interpreted carefully. For example, in the two Lithuanian sites we will measure the number of visits to primary care and the number of consultations over the year-long intervention. One of the aims of the intervention is to reduce hospitalizations and emergency rooms visits and this might mean that primary care visits increase as a result. Therefore, careful interpretation of the significance of a change in primary care visits and what this means in terms of economical and clinical impact is needed. 

The current pilot testing of the Integrated Multimorbidity Care Model in five European sites is expected to provide relevant information that should help in developing integrated care and treatment programs for multimorbid patients. Currently, there is little information in the scientific literature on such interventions [4,20]. Importantly, existing interventions for multimorbid patients frequently apply different methods and care components, making it a challenge to compare strategies. Further, as previous interventions have been applied in a range of counties, with many in the USA [4,20], it is challenging to identify how such models and their components can be applied in different settings, including Europe. Our project aims to bridge that gap by describing integrated care models that have been implemented with a common strategy in different counties and in various care settings (from primary care to outpatient clinics). 

The eventual results of our study may contribute to the knowledge being currently built in other ongoing trials in Europe; specifically two cluster randomized trials in Spain and the UK [17,21]. Results from the UK-based “3D” trial [21] suggest that an intervention on multimorbid patients in primary care consisting of 6-monthly comprehensive reviews did not improve patients’ quality of life or perceived illness burden or treatment burden. Rather than disease focused care, they administered a patient centered care model focused on improving the continuity, coordination, and efficiency of care via comprehensive multidisciplinary assessments twice-yearly. However, although the results of the trial were not successful in terms of improving patient outcomes, it has been noted that the study did not apply patient stratification techniques that might help to identify the most demanding and complex-to-treat groups, who would possibly benefit more from the care intervention [14] (e.g., stratifying according to patterns of different diseases and symptoms or selecting according to specific physical, cognitive, or socioeconomic factors). Other trials, such as the Spanish Multi-PAP project [17] may provide more insight; this study selects patients on the basis of both multimorbidity and polypharmacy. The trial aims to examine the effectiveness of an intervention for improving drug prescriptions and includes an intervention consisting of family-physician training (including concepts such as multimorbidity, appropriate prescribing, and shared decision making) as well as a physician-patient interview based on Ariadne principles (a model of care based on comprehensive assessment that takes into account also the context and preferences of the patient). The Multi-PAPtrial is one of the few studies of this kind that specifically investigates a training element for healthcare professionals and thus these results, together with those from our pilot sites, will provide a more thorough picture of the elements of integrated care that can lead to successful outcomes in multimorbid patients.

In conclusion, the five Integrated Multimorbidity Care Models described in the current paper are currently in the implementation phase and most are due to last for 12 months. Analysis of the common key indicators at the post-implementation phase will help to demonstrate the clinical effectiveness of the care models as well as highlight any difficulties in adapting a common care model in different countries and clinical settings. This information should help to guide future healthcare institutions who wish to implement an Integrated Multimorbidity Care Model in their clinical setting.

## Figures and Tables

**Figure 1 ijerph-16-05044-f001:**
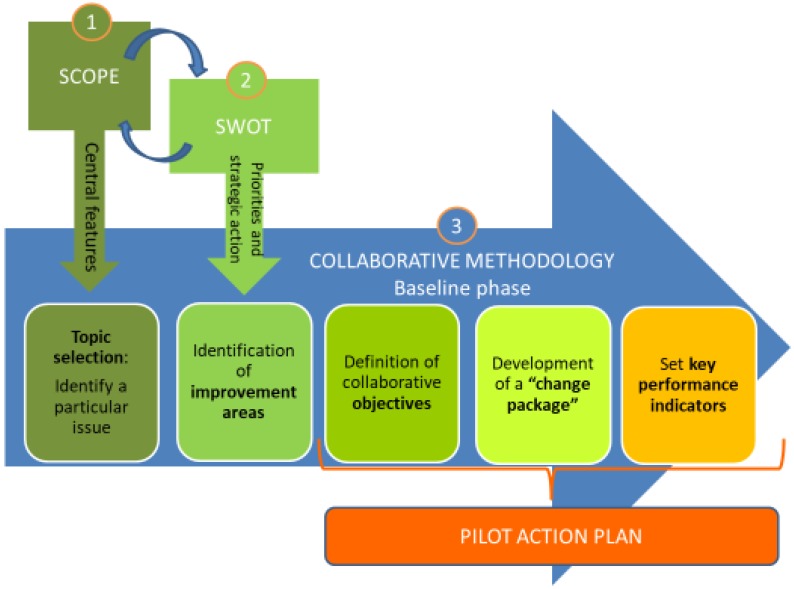
Description of the implementation phases conducted by the local implementation working groups.

**Table 1 ijerph-16-05044-t001:** Local implementation working groups; core set of participants and their relevant tasks and responsibilities.

Participants
Organizer ○ Plan, prepare, chair and run the group workshops ○ Run the secretariat (prepare agendas and minutes) ○ Write reports
Experts ○ Provide knowledge and faculty on specific matters depending on the intervention selected
Decision makers ○ Provide strategic vision ○ Support and sponsorship of the implementation process ○ Eliminate bottlenecks during the implementation process
Front-line stakeholders ○ Give knowledge and expertise on real-life practice experience ○ Choose the right type of subject to implement ○ Motivate and empower implementers ○ Equip and support implementers to deal with the implementation
Implementers (can be same individuals as the front-line stakeholders) ○ Implement the intervention following the agreed plan ○ Continuously assess the implementation process ○ Provide input and feedback to the local implementation group

**Table 2 ijerph-16-05044-t002:** Five steps used to define Action Plans for the Integrated Multimorbidity Care Model.

Action Plan Steps
**1. Identify the specific issues to work on** The central features or elements of the intervention to work were already selected during the definition of the scope. These included components of the Integrated Multimorbidity Care Model [6].
**2. Detect improvement areas** Based on the strengths, weaknesses, opportunities, threats (SWOT) analysis, the working groups identified specific areas for improvement.
**3. Define specific objectives** According to the improvement areas detected, the working groups developed achievable and realistic objectives.
**4. Develop the Change Package** Based on the improvement areas and the associated objectives, concrete activities were described in a “change package”, which is a set of changes that lead to improvement and successful implementation of Integrated Multimorbidity Care Model during the implementation phase. Each objective defined in the previous step requires at least one activity.
**5. Set key performance indicators** Key performance indicators were defined to ensure that the expected impact of the interventions can be accurately measured. Depending on the site, the indicators could either be intermediate health-related outcome measures, process indicators, or both. The targets had to be achievable and measurable. Existing data was chosen to measure progress.

**Table 3 ijerph-16-05044-t003:** Characteristics of the five pilot sites.

	Andalusian Health System	Aragon Health System	UCSC-Rome	VULSK	Kauno Klinikos
**Country**	Spain	Spain	Italy	Lithuania	Lithuania
**Patients**	Complex chronic patients with multimorbidity (patients with chronic severe health problems, multimorbidity and polypharmacy)	Patients with multimorbidity (3+ diseases) and polypharmacy (5+ drugs) or complex	Adults with dementia or Down syndrome and multimorbidity	Patients with multimorbidity (2+ diseases)	Patients with multimorbidity (2+ diseases)
**Age**	No age limit	≥65 years	≥65 years Alzheimer Disease patients ≥18 years Down Syndrome patients	45–70	45–70
**Target number of patients**	All complex chronic patients with individualized care plans initiated from December 2018 to February 2019 all over the region	200	200	200	200
**General aim**	Assess the systematized application of individualized care plans to complex chronic patients	Training of healthcare professionals in multimorbidity + integrated care measures	Improve case coordination, and provide patients with a reference care provider (+Technocare)	To improve the quality of life, decrease the number of potentially avoidable hospitalizations/readmissions and improve quality of multimorbid patient care	To improve the quality of life, decrease the number of potentially avoidable hospitalizations/readmissions and improve quality of multimorbid patient care
**Setting**	Primary care centers in the region	13 primary care health centers +1 hospital of reference	Outpatient clinic	Different primary care health centers (1 public, 1 private)	Different primary care health centers (1 urban, 1 rural)
**Implementation**	All of the five pilot sites include a six-month run-in period (patient recruitment), followed by a 12-month implementation period				

**Table 4 ijerph-16-05044-t004:** Components of the Integrated Multimorbidity Care Model that will be applied in each of the interventions.

	Andalusian Health System ^1^	Aragon Health System	UCSC-Rome	VULSK	Kauno Klinikos
**Delivery of the care model system**					
Regular comprehensive assessment of patients		Yes	Yes	Yes	Yes
Multidisciplinary, coordinated team		Yes	Yes	Yes	Yes
Professional appointed as coordinator of the individualized care plan (“case manager”)		Yes	Yes	Yes	Yes
Individualized care plans	Yes	Yes		Yes	Yes
**Decision support**					
Implementation of evidence-based practice			Yes		Yes
Training members of the multidisciplinary team		Yes		Yes	Yes
Developing a consultation system to consult professional experts		Yes		Yes	Yes
**Self-management support**					
Training of care providers to self-management support		Yes			
Providing options for patients and families to improve their self-management			Yes	Yes	Yes
Shared decision making (care provider and patients)		Yes	Yes	Yes	Yes
**Information systems and technology**					
Electronic patient records and computerized clinical charts		Yes			Yes
Exchange of information between care providers and sectors by clinical information systems		Yes		Yes	
Uniform coding of patients’ health problems where possible			Yes		
Patient-operated technology allowing patients to send information to their care providers			Yes		
**Social and community resources**					
Supporting access to community- and social- resources					Yes
Involvement of social network (informal), including friends, patient associations, family, neighbors			Yes	Yes	

^1^ The Andalusian Health System already has other components of the Integrated Multimorbidity Care Model in place.

**Table 5 ijerph-16-05044-t005:** Specific key performance indicators at five pilot sites that are implementing an Integrated Multimorbidity Care Model: intermediate health-related outcome measures and process indicators.

*Andalusian Health System*	*Aragon Health System*	UCSC-Rome	*Kauno Klinikos* and VULSK
**PROCESS INDICATORS** Number of health districts participating in the pilot Drawing up and delivering the ndividualized care plans Number of primary care units involved Number of visits of complex chronic patients with individualized care plans to primary healthcare centers in 12 months Number of health care professional team meetings related to individualized care plans in 12 months Quality of performed individualized care plans	**PROCESS INDICATORS** Existence of a document describing the functions/role of the case manager Percentage of patients included in the program with case manager identified Number of primary care teams included in the program Implementation of a chronic care unit at the hospital Identification of personnel of reference at hospital’s chronic care unit Number of health professionals who accept to do/start/finish the training course Improvement of knowledge and skills in multimorbidity after the training course Existence of a module of information shared among professionals in the electronic health records Percentage of response to inter-consultations in less than 96 h Availability of direct and specific communication channels between chronic patients and their case managers Social support/needs assessment Identification and mapping of community assets	**PROCESS INDICATORS** A survey will be administered in the outpatient context at the start of the quality improvement intervention and 10 months after the rollout process Reduction of unnecessary referrals Percentage of dropouts (number of missing appointments by patients with AD and DS/number of fixed appointments for patients with AD and DS) calculated as an index for poor coordination of care Average number of Technocare contacts recorded in 12 months Percentage of extra Technocare contacts for Lazio region Percentage of Technocare dropouts (percentage of patients with AD and DS who disattend the fixed Technocare appointment/number of patients with AD and DS who fixed Technocare appointment) Percentage of rescheduled techno visits (percentage of rescheduled visits for patients with AD and DS/number of patients with AD and DS who fixed Technocare appointment Number of patients with AD and DS that participate in the group meeting	**PROCESS INDICATORS** Existence of a guidelines that describes the role of case manager % of patients with individualized care plan based on a comprehensive assessment Number of visits to primary care team in 12 months per patient Number of consultations in 12 months
**IMMEDIATE HEALTH-RELATED OUTCOMES** ACIC and PACIC+ Inpatient episodes of complex chronic patients with individualized care plans in 12 months Outpatient visits of complex chronic patients with individualized care plans in 12 months Emergency episodes of complex chronic patients with individualized care plans in 12 months Rate of unplanned hospitalization potentially preventable achieved in 12 months	**IMMEDIATE HEALTH-RELATED OUTCOMES** ACIC Number of admissions to the emergency room in 12 months Number of hospitalizations in 12 months Number of hospitalizations at chronic care unit/total hospitalizations Satisfaction of the training course by health professionals and self-perceived applicability in clinical practice Percentage of inter-consultations Percentage of patients with individualized care plan based on a comprehensive assessment Prevalence of polymedicated and hyper-polymedicated patients	**IMMEDIATE HEALTH-RELATED OUTCOMES** ACIC and PACIC+ Reduction of accessibility in Emergency Department and subsequent hospitalizations	**IMMEDIATE HEALTH-RELATED OUTCOMES** ACIC and PACIC+ Number of unplanned visits in 12 months Number and duration of hospitalizations, admissions to emergency room, and avoidable hospitalizations in 12 months Number of incompatible drugs combination (drug interaction rate) -EQ-5D questionnaire is a standardized instrument developed by the EuroQol Group as a measure of health-related quality of life -The EQ VAS records the patient’s self-rated health on a vertical visual analogue scale

AD: Alzheimer disease, DS: Down Syndrome, PACIC+: The Patient Assessment of Care for Chronic Conditions+, ACIC: Assessment of Chronic Illness Care questionnaire, EQ VAS: EuroQol-visual analogue scales, EQ-5D: EuroQol 5D.

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
