# Peer review of "A Methodological Approach for Implementing an Integrated Multimorbidity Care Model: Results from the Pre-Implementation Stage of Joint Action CHRODIS-PLUS"

_ijerph, 2019, doi:10.3390/ijerph16245044_

Round 1

Reviewer 1 Report

The Authors developed a practice-oriented implementation methodology for integrated multi-morbidity care models and tested its applicability in five pilot regions in Spain, Lithuania, and Italy. The methodology is clearly written and reproducible in other settings. However, I identified important limitations of the current version and suggest some revisions to the manuscript as listed below.

In the planning phase, Local Implementation Working Group members selected the key features and elements of the five new integrated care models "according to their health context and local needs, interests, and capabilities" (page 6, lines 3-5). This description suggest a solely opinion-based selection and decision mechanism, and it is unclear how the existing evidence from previously implemented IC models was identified and channeled into the decision process. Please describe in the methods section whether the available scientific evidence on effectiveness and cost effectiveness of the selected model elements was systematically reviewed and discussed before the decision, or not. Please also clarify how the relevance/transferability of existing scientific evidence to the specific pilot regions / contexts was assessed. In the discussion section, please also include any corresponding limitation of the applied methods.   An important asset of the conducted research is the systematic identification of internal and external barriers and facilitators of the five local IC models (SWOT analyses). However, the SWOT results are not included in the manuscript at present. Please include the swot findings for all five local models in the manuscript or in the supplementary online materials, since these situation analyses will probably be inspiring for future multi-morbidity IC model planning teams.   Please clarify the planned duration of the models ("6-month run-in period, followed by 12-month implementation period" - Table 3). Patient enrollment will end at 6 months, to allow 12-month follow-up of key performance indicators? What is the rationale to stop the 5 models after 18 months? Do the pilot sites plan the continuation of the models in case of positive performance findings, or not?  In Table 2, Items 2 and 3 have sentence fragments, please check.  Case managers have a key role in IC models - please describe their educational and organizational background, and their corresponding workload (e.g. number of case managers and their work-hours per case) in the model descriptions.  The number of visits to primary care team in 12 months per patient, and the number of consultations in 12 months are two key performance indicators both at Kauno Klinikos and at VULKS. Please explain whether an increase or a decrease is considered here as an improvement - given that these models also aim to reduce the number of unplanned visits, the number and duration of hospitalizations, admission to emergency rooms, and avoidable hospitalizations in 12 months. Reducing the volume of primary care visits may be counterproductive regarding the latter, economically probably much more significant objectives. 

Reviewer 2 Report

Congratulate the researchers for multimorbidity care model that is being implemented. A series of modifications to the manuscript would be necessary.
1. Clarify concepts in the introduction: What is CHRODIS-PLUS? What is Wager's chronic care model?
2. In methodology, the acronym SCOPE and SWOT analysis must be defined.
3. It would be advisable to deeper into discussion and provide more references, are there limitations?
4. A section with conclusions should be provided.

Round 2

Reviewer 1 Report

Thank you for the manuscript adaptations and Author responses, I generally concur with the revised version. However, I tend to disagree with your summary of the current level of evidence, and how to use this evidence in the model planning phase. The manuscript states that "In 2016 a systematic review (10) found only nineteen publications in the scientific literature that assessed integrated care models for multimorbidity, and only one of these was from Europe." And the response document says that "There is little data available and no cost effectiveness studies, because the 2017 Integrated Multimorbidity Care Model has not yet been tested in clinical practice. Importantly, only one study has assessed integrated care components specifically in multimorbid patients in Europe. ..."

Although there are specific challenges to assess effectiveness and cost-effectiveness of integrated care models (as discussed by Tsiachristas et al. in Int J Integr Care. 2016 Oct-Dec; 16(4): 3.), the relevant scientific evidence is apparently broader than a single study in the EU, see e.g. a more recent review of integrated care models for complex elderly patients by Marino et al. (Int J Care Coord 2018,(https://doi.org/10.1177/2053434518817019). However, even in this review, a perfect match to the discussed five models is obviously missing.

Given that all integrated care models are complex and unique in some way, evidence based health policy is either irrelevant in the integrated care model planning and initiation phase, or have to rely on the transferability assessment of existing second-best evidence on other integrated care models with similar key care and cost components in other countries/regions. When developing and promoting an implementation strategy for new integrated care models, the latter approach should be preferred and supported by more practice-oriented recommendations in my view. This is especially true for integrated care models designed for long-term operation and targeting a large number of target patients - unlike the five test models with intentionally limited patient enrollment and duration, adjusted to the secured EU funding in the 3-year JA-CHRODIS PLUS project. 

I look forward reading your follow-up paper on local SWOT findings and more detailed description of the five models.